# Differences in Volatile Profiles and Sensory Characteristics in Plum Spirits on a Production Scale

Josef Balak [1], Lucie Drábová [2], Olga Maťátková [1], Marek Doležal [2], Dominik Marsík [1] and Irena Jarosova Kolouchova [1],*

[1] Department of Biotechnology, University of Chemistry and Technology, Technická 5, 166 28 Prague, Czech Republic; josef.balak@vscht.cz (J.B.); olga.matatkova@vscht.cz (O.M.); dominik.marsik@vscht.cz (D.M.)

[2] Department of of Food Analysis and Nutrition, University of Chemistry and Technology, Technická 5, 166 28 Prague, Czech Republic; lucie.drabova@vscht.cz (L.D.); marek.dolezal@vscht.cz (M.D.)

\* Correspondence: irena.kolouchova@vscht.cz; Tel.: +420-220-444-367

**Abstract:** The specific sensory properties attributed to distillates from different plum varieties are intricately linked to aromatic substances, fruit quality, and technology employed during production. This study compares four plum brandies, each made from a renowned plum variety: Presenta, Valjevka, Čačanská lepotica, and Čačanská rodná on a production scale. Analytical and sensory profiles were assessed using GC-FID, an available analytical method advantageous for monitoring industrial fruit distillate production. Between 71 and 85 analytes were detected in the distillates, with the Presenta plum distillate containing the highest number of substances. Statistically significant differences in analyte concentration between plum varieties ($p < 0.05$) were observed for 11 analytes. The comparison of analytical profiles and sensory evaluation revealed that a high concentration of 1-propanol, despite its negative sensory perception, significantly impacts the overall perception of a distillate, contrasting with substances like acetaldehyde and propyl acetate, which have positive sensory evaluations but lesser significance in content. Our work identified key compounds and procedures that can be used as benchmarks for production of plum brandy with positive sensory evaluation. These findings demonstrate the broad application potential of GC-FID in fruit distillate production as an independent tool for aromatic profile assessment and quality control.

**Keywords:** volatile compounds; plum spirit; sensory characteristic; gas chromatography





## 1. Introduction

Fruit spirits production relies on the conversion of fermentable sugars into ethanol by microorganisms, primarily yeast, followed by distillation. The concentrations of volatile substances, aroma attributes, and olfactory thresholds are critical factors in determining the quality and character of each distillate [1]. The composition and condition of the source fruit play a pivotal role in the sensory attributes and the organoleptic characteristics of the final product, as most compounds are formed during fermentation through microbial activity [2–4]. Technologically important substances are mainly present in the pulp and peel of the fruit, while some sensory active and potentially harmful substances originate in the fruit stones [5,6]. Key groups of substances include saccharides, organic acids, nitrogenous compounds, polyphenols, vitamins, minerals, and volatile fragrances [2]. Pectin breakdown during fermentation, particularly in pectin-rich raw materials, contributes to the presence of methanol in fruit spirits [7]. Methanol and its metabolites present toxicity risks, prompting regulatory limits on their concentration (Regulation (EU) 2019/787 of the European Parliament and of the Council) [8]. The profile of higher alcohols produced is influenced by various factors, notably the range of available amino acids [9–11].

Aromatic compounds, also known as fragrances, constitute a diverse group of volatile compounds that are sensorially active. Many of these compounds traverse the entire

production process and contribute to the desirable fruity aroma of the final distillate. Typically, aromatic volatiles from fruit are found in distillates at lower concentrations, compared to many fermentation products, such as higher alcohols [12,13]. However, their low olfactory thresholds and synergistic effects significantly influence the organoleptic properties of distillates [14–16]. Odorants are found in fruit, either in free or bound form, and include mainly various alcohols, carbonyl compounds, organic acids, esters, and terpene compounds, as well as lactones, phenols, alkanes, and others [3,17,18]. The precursors for the biosynthesis of these aromatic compounds are terpenoids, lipids, amino acids, and carotenoids, from which their primary carbon skeletons are formed. These are subsequently modified by various metabolic routes, mainly by oxidation, reduction, acylation, methylation, or cyclization, resulting in a diverse range of analytical profiles of fragrant volatiles from various fruits [3,19].

Detailed information on the composition of the selected fruit source and the organoleptic properties of individual compounds can help in predicting the quality of the final distillate [1,5,6,20,21]. Although individual varieties of a given fruit species may vary in composition, these differences are generally less significant than those observed between different fruit types [18,21].

Distillates are most often produced from locally abundant raw materials. In Central Europe, distillates are often made from plums, apricots, apples, pears, and grapes (cognac, brandy, etc.). The plum is a popular fruit crop, especially in Europe, where it is grown on 500,000 ha, with a production of around 3 million tons. In the past 20 years, 170 plum cultivars have been developed to adapt to changing climatic conditions, exhibiting characteristics such as disease resistance, a shorter growing season, late flowering, and frost resistance [22,23]. Europe's largest plum producers include Romania, Serbia, France, and Poland. Plums are cultivated for various purposes, including fresh consumption, dried fruit production, and a significant portion of the plum crop is utilized by the distillery industry [24]. In terms of area, plum orchards rank second only to apple trees among all pome and stone fruit trees [25]. The pre-tax turnover for the distillation industry in the EU is estimated to be 26.5 billion euros [26]. The production of plum spirits has a long tradition in many areas of Central, Eastern, and South-Eastern Europe, among other regions. The compounds 1-hexanol, *cis*-3-hexenol, hexanal, nonanal, benzaldehyde, linalool, hexyl acetate, ethyl 2-methylbutanoate, ethyl nonanoate, methyl cinnamate, ethyl cinnamate, γ-octalactone, γ-decalactone, and others are usually associated with plum aromas [5,6,18,27,28].

Studies have primarily focused on the differences in the composition of volatile compounds in plum brandies, but mostly on a laboratory scale, with total sample sizes typically not exceeding 100 L [13,23,25,29–33]. Other works focused on advanced analytical methods such as GC–HRMS (gas chromatography/high-resolution mass spectrometry), HS-SPME (Headspace solid-phase microextraction), GCxGC–TOF-MS (comprehensive gas chromatography time of flight mass spectrometry), SPME–GC–MS (solid-phase microextraction coupled to gas chromatography–mass spectrometry), and in-line conductivity measurements [13,25,34]. To our knowledge, our study stands out for its unique production scale data source and the utilization of available instruments, such as GC-FID. This method can help the producer verify the quality of the distillate, even during the process. We conducted a comparison between the analytical profiles of four plum brandies fermented and distilled on a production scale, alongside their sensory analysis. We identified compounds that occurred frequently and which were correlated with less favorable sensory outcomes. Moreover, compounds preferred by consumers as characteristic of plum brandies were identified. The results show that this method can be used to identify plum brandies with high commercial potential during production scale distillation.

## 2. Materials and Methods

### 2.1. Samples of Plum Brandy

The distillates compared were made from plums of varieties Čačanská lepotica, Čačanská rodná, Valjevka, and Presenta. The fruit of these four varieties were grown in Moravia Region of Czech Republic under similar climatic conditions. The fermentation of each variety was conducted in a 4 t vertical stainless steel tank, with a total content of 3.5 t of plums disintegrated by toothed roller crusher (without crushing the fruit stones). Alcoholic fermentation was carried out in closed vessels at $20 \pm 2\,^{\circ}$C under anaerobic conditions in an air-conditioned fermentation hall. Fermentation was carried out by the natural autochthonous microorganisms. The duration of alcoholic fermentation was 2 months. The distillation of all the plum varieties was conducted using a single pot still with a distillation column (volume 600 L), heated by a gas burner. The process took place one month after the end of fermentation. Distillate samples were collected as average samples from the collected distillate without standardization of the ethanol content. For analysis, samples were standardized to 40% (*v/v*).

### 2.2. Chemicals and Reagents

All compounds were purchased from Sigma-Aldrich Ltd. (Taufkirchen, Germany) and were of at least p.a. purity or higher. This included the following: methyl acetate, ethyl acetate, ethyl butyrate, octyl acetate, ethyl benzoate, propyl benzoate, hexyl hexanoate, isoamyl benzoate, isoamyl decanoate, ethyl hexanoate, 2-phenylethyl octanoate, methyl cinnamate, ethyl cinnamate, ethyl undecanoate, ethyl (E)-2-decenoate, ethyl myristate, methyl palmitate, ethyl oleate, ethyl linoleate, ethyl salicylate, diethyl succinate, methanol, ethanol, 1-propanol, isopropyl alcohol, 1-butanol, 2-butanol, isobutyl alcohol, 2-methyl butanol, isoamyl alcohol, 1-pentanol, 1-hexanol, 1-heptanol, 1-octanol, 1-nonanol, 1-decanol, benzyl alcohol, 2-phenylethanol, 3-ethoxypropionaldehyde diethyl acetal, diethyl acetal, eugenol, geraniol, linalool, citronellol, $\alpha$-terpineol, acetone, hexanal, heptanal, farnesene, nonanal, isovaleraldehyde, furfural, benzaldehyde, 2,4-butanedione, damascenone, and ethyl carbamate. Water purified by a Milli-Q® Integral system supplied by Merck (Darmstadt, Germany) was used throughout the study.

### 2.3. Instruments

For the identification of the distillate components, the gas chromatograph Agilent 6890N, coupled to the 5975 mass spectrometer (GC-MS, Agilent Technologies, Palo Alto, CA, USA), was employed. For the separation of the target analytes, an HP-INNOWAX column (30 m $\times$ 0.250 mm i.d., 0.25 μm film thickness) obtained from Agilent Technologies (Palo Alto, CA, USA) was used. The GC conditions were as follows: oven temperature program: $40\,^{\circ}$C (3 min); $5\,^{\circ}$C·min$^{-1}$ to $240\,^{\circ}$C (5 min); carrier gas helium with a flow rate of 1 mL·min$^{-1}$; injection mode—split (50:1); injection volume 1 μL; inlet temperature $250\,^{\circ}$C. The MSD interface temperature was set to $230\,^{\circ}$C, the quadrupole temperature to $150\,^{\circ}$C and the ion source temperature to $230\,^{\circ}$C. The mass spectrometer was operated in the selected ion-monitoring mode, detecting at least two ions per analyte. The quantification of individual compounds was carried out using an Agilent 7890B gas chromatograph, coupled to a flame ionization detector (GC-FID, Agilent Technologies, Palo Alto, CA, USA). Target analytes were separated using an HP-INNOWAX column (60 m $\times$ 0.250 mm i.d., 0.25 μm film thickness) under the following conditions: $35\,^{\circ}$C (8 min); $5\,^{\circ}$C·min$^{-1}$ to $250\,^{\circ}$C (5 min). The carrier gas was helium, with a flow rate of 1.7 mL·min$^{-1}$; injection mode: split 1:50 with an injection volume of 1 μL. Analyses were carried out in triplicate and their averages were used as a single data point. Calibration was performed and repeated with a standard solution, according to Spaho et al. [11].

### 2.4. Sensory Evaluation

The results of the chemical analyses were supplemented with a sensory analysis. This entailed a consumer test with untrained evaluators who, although not formally trained,

were accustomed to regularly consuming this type of beverage. The samples were evaluated in the laboratory of sensory analysis at the University of Chemistry and Technology, Prague, Department of Food Analysis and Nutrition with 12 boxes, which is equipped according to the relevant international standard ISO 8589 [35]. The procedures of all sensory analyses were in accordance with the international ISO standards. All samples were tested by sensory profile assessment, according to ISO 13299 [36], by a total of 60 people. A 100-point unstructured scale with 12 descriptors was used for the quantitative description of aroma: harmony, delicacy, overall pleasantness (0, unpleasant–100, very pleasant), intensity of the fruity, herbal, spicy, resinous, bitter almond, pungent (sour), technical (chemical) and negative fragrance (0, not noticeable–100, very strong) and overall impression (0, very poor–100, exceptional), as well as by the following 11 descriptors used for the evaluation of flavor: harmony, delicacy, aftertaste, overall pleasantness (0, unpleasant–100, very pleasant), intensity of the fruity, bitter, sour, alcoholic and negative aftertaste (0, not noticeable–100, very strong) and overall impression (0, very poor–100, exceptional). RedJade software (RedJade Sensory Solutions, LLC, Martinez, CA, USA) was used for the collection of sensory analysis data and their processing.

### 2.5. Statistical Analysis

Statistical analysis was performed using MetaboAnalyst 6.0 software, where a one-way analysis of variance (ANOVA) with Tukey's post hoc test was used for the comparison of the brandy composition. Dixon's Q test was used for the detection of outliers in data obtained (the analysis of each sample was performed in three parallels, the deviation of the five three analyses was less than 5%).

## 3. Results

### 3.1. Analytical Profile

Numerous volatile substances detected in fruit distillates are regulated by EU legislation [7,37]. As per this decree, fruit spirits are classified as spirits with a minimum alcohol content of 37.5% (vol.), and their distillation is conducted to achieve an alcohol content of less than 86% vol. No pure ethanol or even distillate of other agricultural origin may be added to fruit distillates. Fruit distillates contain at least 200 g of volatile substances per hectoliter of 100% vol. ethanol. The addition of flavoring substances is not permitted. Legislation also defines the limits on the content of some substances toxic to the human organism, which could potentially occur in undesirable concentrations in the distillate. Distillates produced from pitted fruit must not exceed a hydrocyanic acid content of 7 g/hL of 100% vol. ethanol. The methanol content for distillate made from plums is regulated to a maximum of 12 g/La (methanol per liter of 100% vol. ethanol), according to this regulation. Fruit distillates may contain ethyl carbamate, which is potentially mutagenic for the human body. Its content in fruit distillates is regulated by Commission Recommendation (EU) 22/2016 [38], which states the recommended maximum concentration of ethyl carbamate as 1 mg/L. The contents of methanol and total volatile substances were significantly lower in all varietal plum brandies in this study.

To observe the influence of the plum variety on the analytical profile of fruit distillates, distillates from plum varieties Presenta, Valjevka, Čačanská lepotica, and Čačanská rodná were selected. Depending on the plum variety, between 71 and 85 analytes were detected in the distillates. The highest number of substances was detected in the sample of Presenta plum distillate. Table 1 lists the most important substances affecting the resulting aroma. An ANOVA was used to assess the significance of difference between plum varieties, which is presented in Table 1 by $p$-value. As can be seen from Table 1, a statistically significant difference in analyte concentration between plum varieties ($p < 0.05$) was detected for 11 analytes. A table with the quantification of the listed compounds is in the Supplementary Materials (Table S1). The sum of higher alcohol concentrations were, for individual varieties, as follows: Presenta 538.16 mg/La, Valjevka 1188 mg/La, Čačanská lepotica 570 mg/La, and Čačanská rodná 4098 mg/La.

**Table 1.** Differences in the analyte concentration among the different plum varieties determined as *p*-value and ANOVA test. Significant difference: *p* < 0.05. * Statistically significant differences in analyte concentration.

| Major Compounds | RT (min) | *p* | Minor Esters | RT (min) | *p* |
|---|---|---|---|---|---|
| Methyl acetate | 4.813 | 0.678 | Methyl acetate | 4.813 | 0.732 |
| Ethyl acetate | 6.019 | 0.725 | Propyl acetate | 9.29 | 0.047 * |
| Methanol | 6.305 | 0.341 | Butyl acetate | 13.509 | 0.824 |
| 2-Butanol | 11.47 | 0.379 | Isoamyl acetate | 15.257 | 0.052 |
| 1-Propanol | 12.058 | 0.013 * | Ethyl caprylate | 25.167 | 0.240 |
| Isobutanol | 14.305 | 0.041 * | Ethyl caprate | 30.258 | 0.259 |
| Isoamyl alcohol | 18.404 | 0.237 | Ethyl benzoate | 31.092 | 0.026 * |
| Ethyl (-)-L-lactate | 22.692 | 0.240 | Diethyl succinate | 31.224 | 0.021 * |
| Acetic acid | 25.590 | 0.374 | Ethyl (*E*)-2-decenoate | 33.136 | 0.065 |
| Minor alcohols | | | Ethyl salicylate | 34.352 | 0.034 * |
| 1-Butanol | 16.216 | 0.283 | 2-Phenylethyl acetate | 34.407 | 0.053 |
| 1-Pentanol | 18.404 | 0.233 | Ethyl laurate | 34.82 | 0.238 |
| 1-Hexanol | 22.88 | 0.605 | Ethyl myristate | 38.975 | 0.082 |
| Linalool | 28.034 | 0.620 | Ethyl cinnamate | 40.744 | 0.214 |
| 1-Octanol | 28.282 | 0.045 * | Ethyl palmitate | 42.794 | 0.058 |
| 1-Nonanol | 30.723 | 0.0004 * | Ethyl stearate | 46.322 | 0.027 * |
| 1-Decanol | 33.034 | 0.001 * | Ethyl oleate | 46.717 | 0.336 |
| Nerol | 33.949 | 0.298 | Ethyl linoleate | 47.478 | 0.086 |
| Geraniol | 34.923 | 0.098 | Carbonyl compounds | | |
| Benzyl alcohol | 35.661 | 0.118 | Acetaldehyde | 3.538 | 0.019 * |
| 2-Phenylethanol | 36.392 | 0.181 | Acetone | 4.615 | 0.144 |
| Eugenol | 41.381 | 0.107 | 2,3-Butandione | 9.591 | 0.365 |
| | | | Furfural | 26.135 | 0.298 |
| | | | Benzaldehyde | 27.682 | 0.283 |

Figures 1–4 show the mean relative substance context (%) of analytical profiles for the different groups of substances in the four varietal plum brandies. The concentrations of many compounds varied significantly among the samples. The plum brandy made from Presenta plums, was characterized by the highest concentrations of methyl acetate, ethyl acetate, methanol (Figure 1), 1-pentanol, 1-nonanol, geraniol, and benzyl alcohol among the samples (Figure 4). The sample of plum brandy made from the variety Valjevka had a moderately high content of 1-propanol (Figure 1) and, on the contrary, it had the lowest contents of acetaldehyde, 2,3-butanedione, and also other carbonyl compounds (Figure 2). The plum brandy produced from the Čačanská lepotica plum variety contained the highest concentrations of isobutyl alcohol, isoamyl alcohol, 1-decanol, acetone, 2,3-butanedione, benzaldehyde, ethyl decanoate and ethyl laurate compared to the other samples, and, on the other hand, the lowest concentrations of methyl and ethyl acetate, 1-propanol, acetic acid, linalool, benzoic acid, isoamyl acetate, ethyl benzene, ethyl palmitate, and ethyl oleate. The brandy produced from plums of the Čačanská rodná variety contained very high concentrations of 1-propanol, acetaldehyde, propyl acetate, isoamyl acetate and linalool. Compared to the other samples, it contained the lowest concentrations of methanol, 1-butanol, geraniol, ethyl octanoate, ethyl decanoate, ethyl laurate, ethyl myristate and ethyl linoleate.

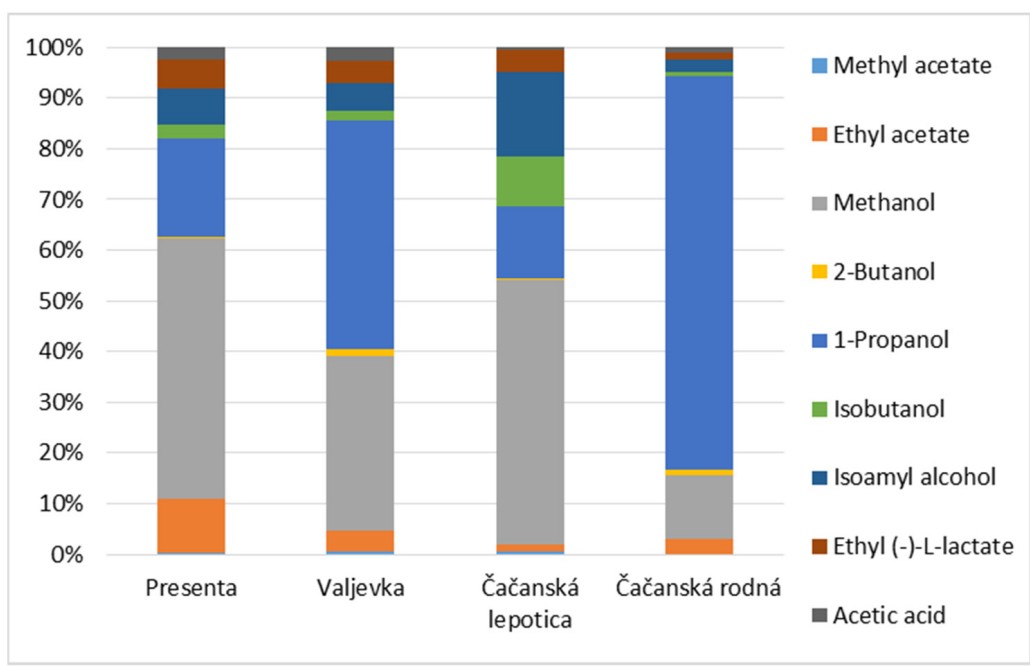

**Figure 1.** Mean relative contribution (%) of major compounds in plum brandies produced from Presenta, Valjevka, Čačanská lepotica, and Čačanská rodná plum varieties.

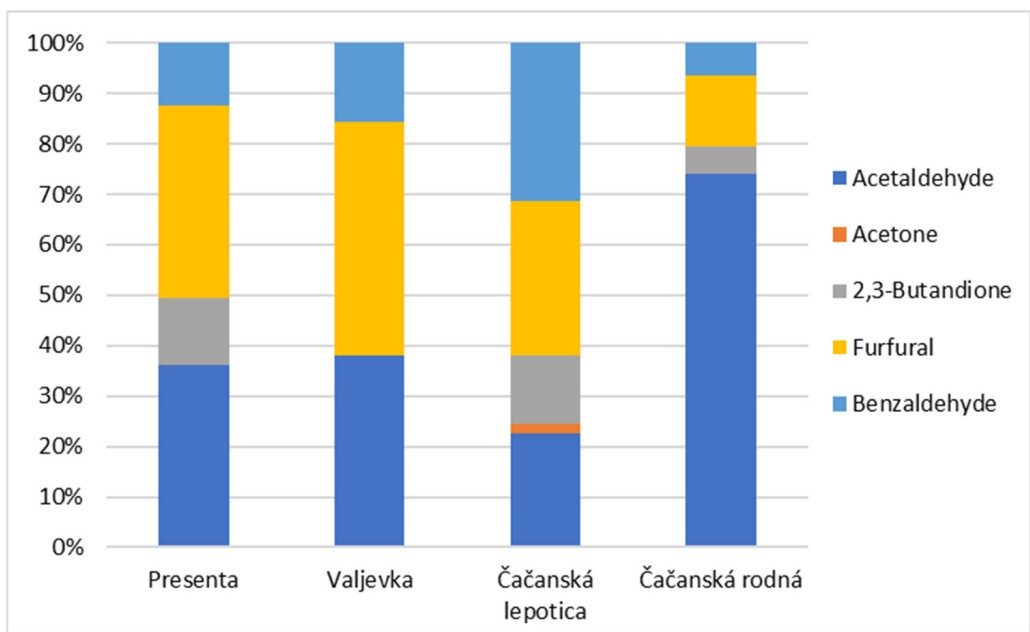

**Figure 2.** Mean relative contribution (%) of carbonyl compounds in plum brandies produced from Presenta, Valjevka, Čačanská lepotica, and Čačanská rodná plum varieties.

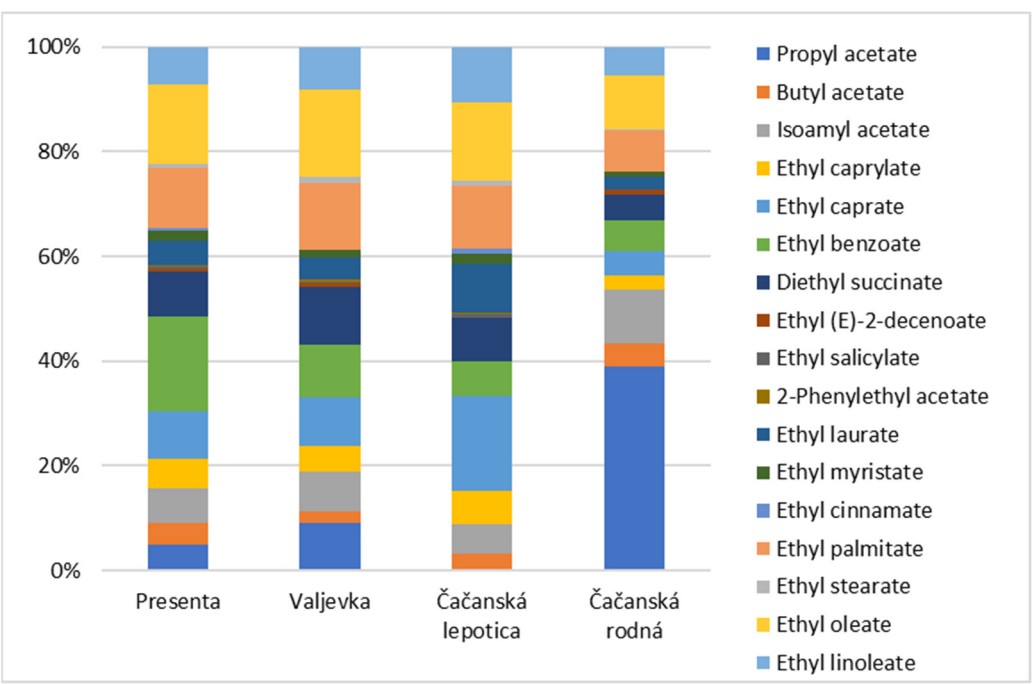

**Figure 3.** Mean relative contribution (%) of minor esters in plum brandies produced from Presenta, Valjevka, Čačanská lepotica, and Čačanská rodná plum varieties.

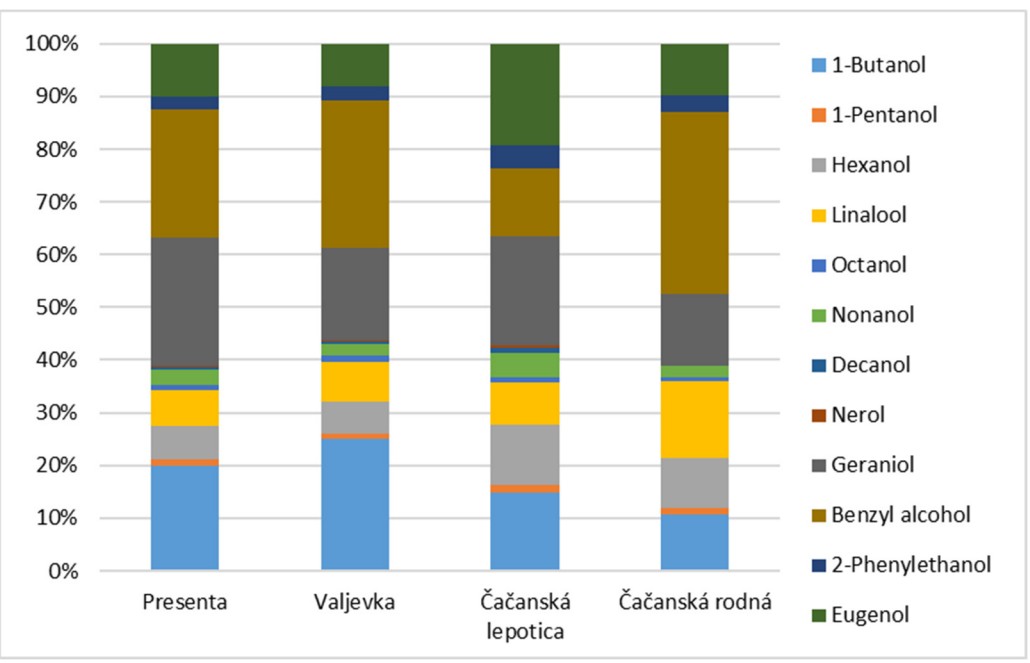

**Figure 4.** Mean relative contribution (%) of minor alcohols in plum brandies produced from Presenta, Valjevka, Čačanská lepotica, and Čačanská rodná plum varieties.

*3.2. Sensory Profiles*

During the sensory evaluation, involving 60 assessors, the sensory profiles of the four studied plum brandies were determined. The samples were ranked according to the personal preferences of the assessors, from best to worst. According to the ranking test, the best-ranked sample was plum brandy made from the Presenta variety (Figure 5). This result was determined at a probability value of $p$ = 98.5%, so it can be assessed as conclusive. A statistically significant difference was found in the intensity of the bitter taste between samples Čačanská rodná and Čačanská lepotica and samples Presenta and Valjevka, which

can be regarded as moderate evidence of a difference, at the established probability level of $p = 97.2\%$. There was established a statistically significant difference in the intensity of the fruit flavor for the Presenta variety (probability level of $p = 96.3\%$). The taste and flavor profiles of these samples are shown in Figures 5 and 6.

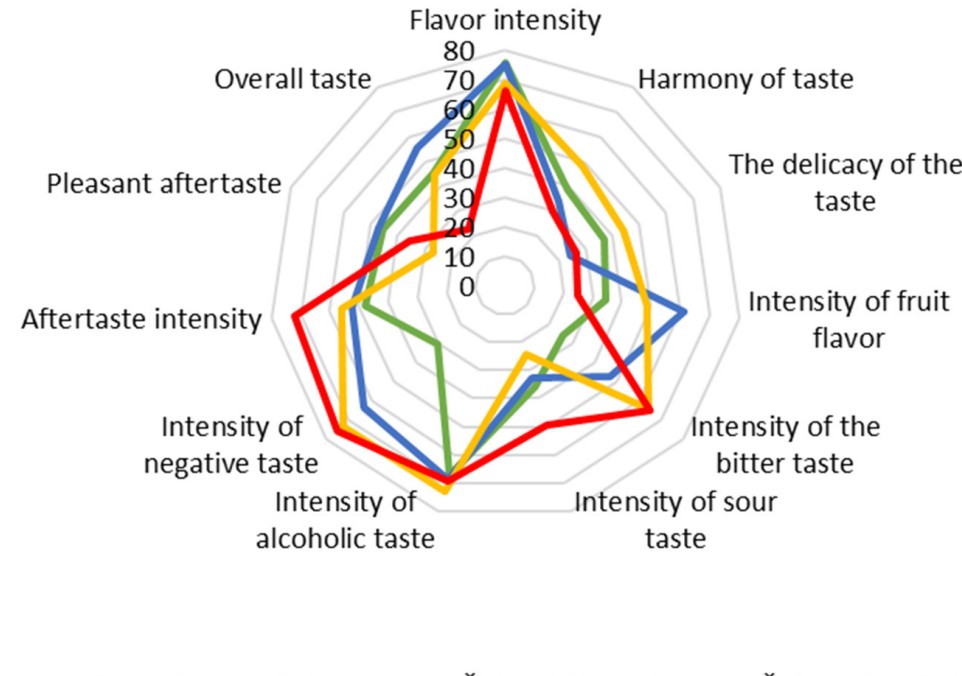

**Figure 5.** Flavor profiles of plum brandies produced from Presenta, Valjevka, Čačanská lepotica, and Čačanská rodná varieties.

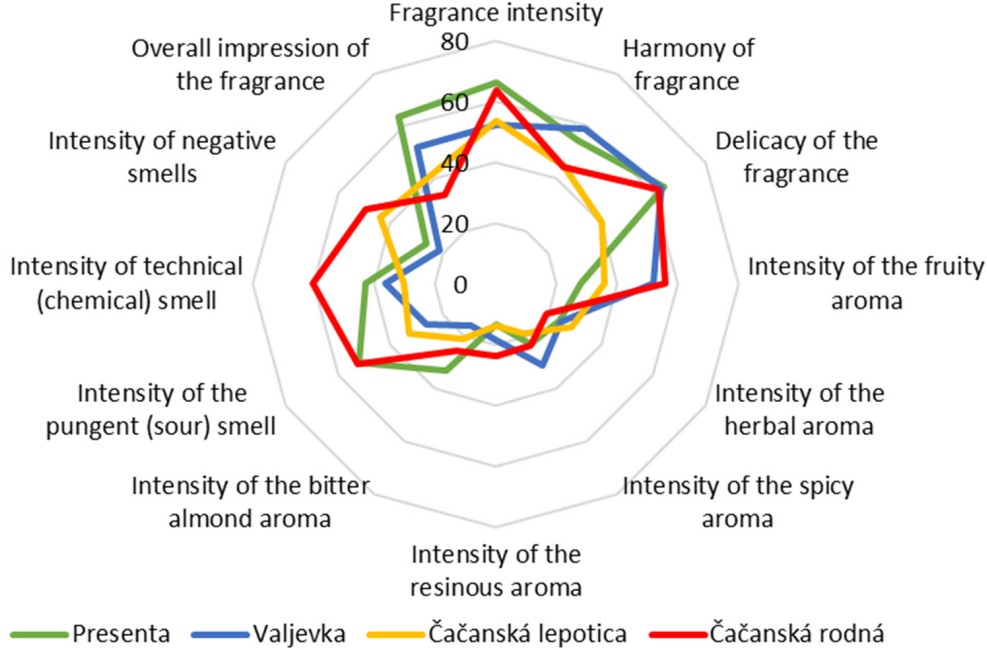

**Figure 6.** Aroma profiles of plum brandies produced from Presenta, Valjevka, Čačanská lepotica, and Čačanská rodná plum varieties.

Based on statistically significant differences in analyte concentration between plum varieties, Čačanská rodná, the worst-rated variety, exhibited differences in the concentrations of five compounds. Here, the highest concentrations were observed for 1-propanol, acetaldehyde, and propyl acetate. Conversely, 1-decanol and ethyl salicylate were not detected. Acetaldehyde and propyl acetate are known to have pleasant, fruity odors. 1-propanol is characterized by a sharp smell. The highest-rated variety Presenta had the highest concentrations for 1-nonanol (fruity odor), diethyl succinate (fruit flavors), and ethyl benzoate (fruit flavors), as well as high concentration of geraniol (rose-like odor) and 1-octanol (waxy type odor). All these substances, which were present in high and statistically significant concentrations in the Presenta variety, contribute to its fruity aroma and taste. Plum brandy of the Valjevka variety was rated as the second best. It had high concentrations of 1-octanol, diethyl succinate, 1-butanol, ethyl stearate, and ethyl benzoate. These are also substances responsible for pleasant, fruity aromas and the smell of wax.

Correlation analysis was used to investigate the correlation between sensory evaluation scores and aroma compounds detected in plum spirits. The results showed that the main indicators of positive sensory evaluation were more significantly related to the esters, terpenes, aldehydes, and ketones than to the other compounds (see Table 2). As can be seen from the table, the perception of flavor and fragrance intensity correlates well with terpene and ester content. On the other hand, a higher content of higher alcohols has a strong influence on the perception of resinous and technical smells, as well as on bitter and negative tastes. A low content of terpenes, aldehydes, and ketones contributed positively to the sensory quality of plum spirits. Generally, many aroma compounds in plum spirits will have synergistic effects on its flavor; an unbalanced aroma compound level would have adverse effects on the sensory quality of the plum spirit.

**Table 2.** Correlation analysis between sensory evaluation scores and aroma compounds in plum spirits.

| Compounds Categories | Fragrance Intensity | Harmony of Fragrance | Delicacy of Fragrance | Intensity of Fruity Aroma | Intensity of Herbal Aroma | Intensity of Spicy Aroma | Intensity of Resinous Aroma | Intensity of Bitter Almond Aroma | Intensity of Pungent Smell | Intensity of Technical Smell | Intensity of Negative smells | Overall Impression of Fragrance |
|---|---|---|---|---|---|---|---|---|---|---|---|---|
| Higher alcohols | 0.355 | −0.474 | 0.342 | 0.776 | −0.864 | 0.077 | 0.945 | 0.011 | 0.466 | 0.899 | 0.612 | −0.723 |
| Total esters | 0.753 | 0.455 | 0.927 | −0.029 | −0.677 | 0.399 | 0.176 | 0.625 | 0.643 | 0.577 | −0.418 | 0.517 |
| Acetic acid | 0.471 | −0.927 | −0.459 | −0.222 | −0.088 | −0.914 | 0.027 | 0.503 | 0.603 | 0.353 | 0.867 | −0.488 |
| Terpenes | 0.610 | 0.420 | 0.383 | −0.768 | 0.049 | −0.086 | −0.611 | 0.791 | 0.472 | −0.057 | −0.525 | 0.873 |
| Aldehydes and ketones | 0.177 | −0.846 | −0.753 | −0.448 | 0.332 | −0.991 | −0.306 | 0.342 | 0.294 | −0.066 | 0.737 | −0.360 |

| Compounds Categories | Flavor Intensity | Harmony of Taste | Delicacy of the Taste | Intensity of Fruit Flavor | Intensity of Bitter Taste | Intensity of Sour Taste | | Intensity of Alcoholic Taste | Intensity of Negative Taste | Aftertaste Intensity | Pleasant Aftertaste | Overall Taste |
|---|---|---|---|---|---|---|---|---|---|---|---|---|
| Higher alcohols | 0.100 | −0.274 | −0.057 | −0.872 | 0.862 | 0.682 | | 0.396 | 0.947 | 0.858 | 0.257 | −0.534 |
| Total esters | 0.908 | 0.219 | 0.076 | −0.082 | −0.090 | 0.639 | | 0.261 | 0.130 | 0.484 | 0.991 | 0.662 |
| Acetic acid | 0.124 | 0.648 | 0.845 | 0.109 | 0.743 | 0.547 | | 0.883 | 0.613 | 0.558 | −0.090 | −0.667 |
| Terpenes | 0.818 | 0.711 | 0.482 | 0.696 | −0.481 | 0.266 | | 0.294 | −0.402 | −0.021 | 0.673 | 0.775 |
| Aldehydes and ketones | −0.112 | 0.660 | 0.814 | 0.393 | 0.468 | 0.174 | | 0.669 | 0.268 | 0.165 | −0.385 | −0.623 |

## 4. Discussion

The selection of suitable raw materials is the most important technological step influencing the quality of fruit brandy. Rotten or damaged fruit can introduce unwanted microorganisms, contaminating the mash [4,39,40]. Ethyl acetate, propyl acetate, fatty acid esters, isoamyl alcohol, 2-methyl-1-butanol, and 2-phenylethanol (Figures 1–4), which are present in fresh plum distillate, are generated during fermentation by the activity of yeasts [1,41–43].

Butanol can also be produced during fermentation by certain microorganisms, such as bacteria of the genus *Clostridium* [44]. In yeast, only endogenous pathways for the production of 1-butanol (Figure 4) have been described. Butanol can therefore, appear in the plum brandy as a result of cell lysis during fermentation and further processing [44].

The presence of benzaldehyde (Figure 2) in plum distillates is due to the enzymatic degradation of its precursors found in the stones of fruit occurring during fermentation [41]. Benzaldehyde is typically found in small amounts in plum pulp; however, it can be converted into benzyl alcohol by yeasts during fermentation [45]. Delfini reports that at low glucose concentrations, wine yeasts metabolize benzyl alcohol to benzoic acid [46]. Thus, the concentration of benzaldehyde and its derivatives can increase during fermentation.

Geraniol (Figure 4), which according to many authors is present in plum brandy [12,47,48], is not found in plums [18,49]. Some fragrances, such as terpene alcohols geraniol, linalool, nerol, and α-terpineol, are typically found bound in fruit in the form of glycosides [50]. El Hadi suggests that these substances are predominantly present in fruit in bound form rather than free form and are released spontaneously during ripening or by the action of low pH, heat, or enzymes [3]. Yeasts have also been shown to produce terpenoids and/or transform them [51]. Thus, geraniol concentration can serve as an indicator of the content of aromatic terpene alcohol precursors in plums. The highest concentration of geraniol was found in the variety Presenta, followed by the variety Valjevka (Table S1), which were also the best-rated.

Compounds detected in the plum brandies can, to some extent, serve as markers of the quality of the plums in relation to their fragrance content. With this aim in mind, the most important compounds selected for analysis were as follows: 1-pentanol, 1-hexanol, 1-octanol, 1-nonanol, linalool, geraniol, nerol, eugenol, and ethyl cinnamate [23,25,52–54]. All of these substances, except 1-pentanol, were present in the samples of varietal plum brandies at concentrations higher than their threshold of olfactory perception in water, and thus probably contribute to the organoleptic properties of these plum brandies and their characteristic fruity aroma [18,55]. 1-Pentanol is characterized by balsamic, almond, and alcoholic odors [56]. Pino and Quijano report its olfactory perception threshold in water as 4 mg/L [18]. This compound could also influence the organoleptic properties of the distillate through a synergistic effect with other fragrant volatiles [15]. Concentration of 1-octanol and 1-nonanol were in statistically significant differences concentration in all four plum varieties (Table 1). When the concentrations of aromatic compounds are expressed relative to the amount of alcohol, plum brandy made from the Presenta variety plums exhibits the highest abundance of all these compounds, compared to the other studied brandies. The Presenta variety also had the statistically highest concentration of substances responsible for the fruity aroma, as described by Popovic [26] and Ivanovic [24]. The plum brandy made from plums of the Čačanská lepotica variety also had a high content of 1-hexanol, 1-nonanol, nerol, and eugenol. Plum brandy made from plums of the Čačanská rodná variety had a significantly high linalool content and a relatively high 1-hexanol content, but at the same time contained low amounts of other fruit volatiles.

The fragrance content is not the only factor associated with the raw material that influences the analytical profile of fruit spirits. Amino acids in plums are precursors for the formation of higher alcohols during fermentation, and their composition in fruits of different varieties can thus influence the analytical profiles of higher alcohols in various distillates [10]. The concentration of higher alcohols was highly variable in the four varietal

distillates (Table S1). Variability in higher alcohol contents has also been presented by other distillates [6,13]. Spaho suggests that the ratios between the contents of certain higher alcohols in plum brandies are characteristic of plums from different varieties, enabling their identification based on these ratios [11]. Higher contents of isobutyl alcohol, isoamyl alcohol, and 2-phenylethanol were found in plum brandy made from plums of the Čačanská lepotica variety compared to the others, which may indicate a high content of nitrogenous precursors of these alcohols in fruit of this variety. The total content of higher alcohols was the highest in the Čačanská rodná variety, where it reached a value of 410 g/hL of 100% vol. ethanol, above the value of 350 g/hL of 100% vol. ethanol, which is stated as the amount required for a negative sensorial assessment [31]. 1-Propanol contributed significantly to the negative sensory evaluation of plum brandy from the Čačanská rodná variety. The concentrations of other sensory active substances with a positive perception are in higher concentrations than the threshold values and often in higher concentrations than Valjevka, which was rated as the second best. The excellent analytical and sensory qualities of the Čačanská rodná variety are also mentioned by Popović et al., 2019 [23]. The intensity of production of higher alcohols also depends on the composition of fermenting microorganisms, and, therefore, the composition of amino acids in the fruit may not be the only determinant for the analytical profile of higher alcohols in the distillates [13,52].

A sample of plum brandy made from plums of the Čačanská rodná variety had very high concentrations of acetaldehyde, 1-propanol, 2-butanol, and propyl acetate, compared to the other varieties. All four varieties were fermented on the same production scale (4 t), so the difference in the composition of the microflora of this variety may be the determining factor [32,57] in the resulting analytical and sensory profile of the product from this variety [1,58]. A high concentration of 1-propanol is generally a sign of contamination, according to Apostolopolou et al., 2005, as this compound may originate from the development of bacteria during the storage phase of the raw material. From the results, the variety Čačanská rodná is apparently overripe, and the data show how it is difficult to ensure optimal maturity and a short storage time on a large scale of production to avoid the development of epiphytic bacterial microflora [59]. Weather conditions in each season have a much greater effect on the yeast community in the fruit. Due to the high content of yeasts other than *Saccharomyces*, the fermentation process of plums must be monitored [32]. As confirmed by correlation analysis (Table 2) and in accordance with the literature [31], high levels of higher alcohols are considered to be indicators of sensory defects in plum brandies. However, apart from the negative effect of 1-propanol on the sensory profile of the distillate from the variety Čačanská rodná, it can be noted that all distillates were characterized by small differences in sensory evaluation. Satora et al., 2017, reached similar conclusions by comparing different varieties of plums [13].

For the further use of operational control using GC-FID, it would be good to identify unknown substances occurring in higher concentrations, which may contribute to the resulting sensory character of the distillate [31]. It is also possible to connect Electronic nose (E-NOSE) [57–59] and convert the detected data again for GC-FID, usable by manufacturers for in-time evaluation of distillation.

## 5. Conclusions

Fruit spirits are a sought-after consumer product. The commercial success of plum brandy hinges on sensory evaluation, which is influenced by the analytical profile of volatile substances. Therefore, there is a need for reliable, independent, and quick analytical methods that can be utilized by major brandy producers, such as GC-FID, during the production process. We found that high values of 1-propanol fundamentally negatively affected the sensory evaluation of the plum brandy. In large operations, it is often not possible to control the optimal maturity of the raw material as well as the immediate fermentation of large volumes without storing the raw material and thus enabling the possible development of bacterial microflora. This further increases the need for the rapid and affordable quantification of problematic compounds. Statistically significant

concentrations of ethyl benzoate, 1-octanol, 1-nonanol, diethyl succinate, and ethyl stearate have a significantly positive effect on the sensory perception of the final product. Hence, when implementing laboratory control on a production scale with GC-FID, the emphasis should be on determining and regulating the concentrations of compounds perceived negatively, rather than focusing solely on the concentrations of fruit markers.

**Supplementary Materials:** The following supporting information can be downloaded at: https://www.mdpi.com/article/10.3390/fermentation10050235/s1, Table S1: Quantification of the listed volatile compounds.

**Author Contributions:** I.J.K.: conceptualization, investigation, supervision, writing—original draft, and writing—review and editing. J.B.: methodology, formal analysis, and writing—original draft preparation. L.D.: software, validation, and data analysis. O.M.: data curation, writing—original draft preparation, and writing—review and editing. M.D.: data analysis. D.M.: writing—original draft preparation and writing—review and editing. All authors have read and agreed to the published version of the manuscript.

**Funding:** This research was funded by specific university research FPBT 2024 067.

**Institutional Review Board Statement:** Not applicable.

**Informed Consent Statement:** Not applicable.

**Data Availability Statement:** Data are available upon reasonable request from the corresponding author.

**Conflicts of Interest:** The authors declare no conflicts of interest.

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
