# Peer review of "Differences in Volatile Profiles and Sensory Characteristics in Plum Spirits on a Production Scale"

_fermentation, doi:10.3390/fermentation10050235_

Round 1

Reviewer 1 Report

Comments and Suggestions for Authors

The acronym GC should not appear in the keywords, as it is written as an acronym. If the authors consider it important to include it, due to the analytical method used, they must write it in full.

In the introduction, the authors briefly mentioned some fruits used in the production of distillates, but did not emphasize the importance of plums in the subject. I suggest you write a little more about the use or importance of this fruit so that there is a link between the title and the objective mentioned.

The authors briefly described how the sensory analysis was carried out, but the parameters of the attributes evaluated were not clear. Such information could only be viewed in the results. I suggest adding aroma and flavor profiles to the methodology.

The discussion paragraph presents the topics in a running manner without referring, at any time, to the tables and graphs presented in the results topic. Therefore, readers cannot follow which part of the text the authors are referring to. I suggest mentioning which result the text will discuss at each time it is discussed.

The conclusion text: " Our work identified 303 key compounds and procedures that can be used as benchmarks for production of plum 304 brandy with positive sensory evaluation. The analysis was performed at production scale 305 (each plum brandy produced at 4t scale) by analytical and sensory evaluation of four plum 306 brandies made from different varieties"  in fact they are objective and methodology, and should not be repeated here. I suggest removing it.

Author Response

Thanks to the reviewer for all the points and comments. We incorporated everything into the manuscript, and we attach the answers to each question and comment.

Reviewer 2 Report

Comments and Suggestions for Authors

This research fails to meet the minimum standards required by the journal and therefore should be rejected. The methodology lacks sufficient detail in the materials and methods section, particularly regarding the consumer study conducted, including the specific test, conditions and methods of data collection. While sensorial and chemical analyses were performed, the discussion lacks any real correlation between the two. It is recommended that Multi Factor Analysis be conducted to identify correlations between aromatic compounds and sensory perception. Furthermore, the manuscript requires a thorough review of the English language, as numerous typos and unclear sentences are present. Below are some examples:

Abstract conclusions are not very clear (lines 19-22).

Sensory test should be specified (JAR, CATA…)

Software version missing (line 136)

Lines 189-193 are repeated from the abstract and do not follow the flow of the manuscript.

Statistically significant differences in analyte concentration should be noted by asterisk and not highlighted (table 1).

Comments on the Quality of English Language

Extensive editing should be done as multiple typos and unclear sentences are found throughout the manuscript.

Author Response

(The authors gave the same response as above.)

Reviewer 3 Report

Comments and Suggestions for Authors

It is an interesting comparative analysis, in attached are some comments.

Author Response

(The authors gave the same response as above.)

Round 2

Reviewer 2 Report

Comments and Suggestions for Authors

Authors did a superb job rewritten the manuscript and including all the missing information. Methodology has been extensively explained and data analysis has been improved. Besides their efforts, there are still some minor issues that need to be addressed prior to publication.

Line 74: Sentence is unfinished.

Line 96: SPME–GC–MS and in-line conductivity measurement.

Line 123: The process took place after one month after end of fermentation. Remove after.

Figure 1: Purple line not described in caption. Define or delete.

Line 333: Benzaldehyde is typically found in small in plum pulp... Small quantities/concentrations/amount

Comments on the Quality of English Language

Minor errors and typos should be addressed.

Author Response

We thank the reviewer for all the comments that we were able to incorporate and improve our manuscript.

Line 74: Sentence is unfinished. - sentence has been edited.

Line 96: SPME–GC–MS and in-line conductivity measurement. - sentence has been edited

Line 123: The process took place after one month after end of fermentation. Remove after. - sentence has been edited

Figure 1: Purple line not described in caption. Define or delete. - the figure has been checked

Line 333: Benzaldehyde is typically found in small in plum pulp... Small quantities/concentrations/amount - sentence has been edited